# Factors Associated with Lifestyle Behaviors among University Students—A Cross-Sectional Study

**DOI:** 10.3390/healthcare12020154

**Published:** 2024-01-09

**Authors:** Shaima A. Alothman, Alia Abdulaziz Al Baiz, Abeer Salman Alzaben, Ruqaiyah Khan, Ali Faris Alamri, Asma B. Omer

**Affiliations:** 1Lifestyle and Health Research, Health Sciences Research Center, Princess Nourah Bint Abdulrahman University, Riyadh 11564, Saudi Arabia; shaalothman@pnu.edu.sa; 2Department of Epidemiology and Biostatistics, Health Sciences Research Center, Princess Nourah Bint Abdulrahman University, Riyadh 11564, Saudi Arabia; aaalbaiz@pnu.edu.sa; 3Department of Health Sciences, College of Health and Rehabilitation Sciences, Princess Nourah Bint Abdulrahman University, Riyadh 11671, Saudi Arabia; asalzaben@pnu.edu.sa; 4Department of Basic Health Sciences, Deanship of Preparatory Year for Health Colleges, Princess Nourah Bint Abdulrahman University, Riyadh 11671, Saudi Arabia; rkrehman@pnu.edu.sa; 5King Abdullah Bin Abdulaziz University Hospital, Princess Nourah Bint Abdulrahman University, Riyadh 13415, Saudi Arabia; afalamri@pnu.edu.sa

**Keywords:** HPLP II, lifestyle behaviors, physical activity, physical disability, sedentary behaviors, sleep behavior

## Abstract

Lifestyle behaviors are daily habits influenced by social and environmental factors. This study examined lifestyle behaviors and their associations with sociodemographics, comorbidities, and pain in Saudi university students during the academic year 2021 and 2022. All students received the study invitation via university emails to complete an online questionnaire. The questionnaire included four sections (sociodemographics, health-related information, desired health promotion activities, and a lifestyle behavior assessment) via Health-Promoting Lifestyle Profile II (HPLP-II). The associations between study variables were assessed using Pearson’s correlation and multiple linear regression. The study questionnaire was completed by 1112 students. No correlation was found between sociodemographics and lifestyle-behavior-related factors except for students in the College of Science who appeared to have good lifestyle behaviors (an increase in HPLP II total scores of 3.69). Students with mental health issues have poorer lifestyle behaviors and spend more time sitting (*p* < 0.00). Students without disabilities have lower scores in health responsibility, physical activity, nutrition, and stress management, while auditory disability specifically lowers health responsibility (*p* < 0.00). Pain was not associated with any assessed lifestyle behaviors. This study identified several significant correlations and differences between variables such as age, sedentary behavior, sleep duration, disability status, college major, and lifestyle behaviors among PNU students. These findings provide insights into the factors that influence students’ health-promoting behaviors and can help guide interventions for promoting healthier lifestyles on campus. Targeted health promotion strategies at an early age could help in decreasing overall noncommunicable disease incidents later in life. The study results should be interpreted taking into consideration that the collected data were cross-sectional and self-reported. In conclusion, the findings of this study clearly demonstrate the need for specific lifestyle and health-promoting programs that are directed toward university students.

## 1. Introduction

Lifestyle behaviors have been defined as daily practices that are determined by different social and environmental factors [1]. The American College of Lifestyle Medicine has determined six pillars of lifestyle medicine, including food and dietary intake, physical activity, sleep, stress management, tobacco use, and social connections [2]. Different populations based on geographical location, age, disease status, or any common grouping factor exhibit different trends and norms of each behavior; thus, it is imperative to assess the above-mentioned lifestyle behaviors and their related associates within each specific population to generate relevant health data.

Lifestyle behaviors have a direct influence on the incidence of comorbidities [3,4]. Comorbidities can be defined using different concepts. However, the most accepted definition of comorbidities is the physiological and/or psychological burden of a medical condition on the individual [5]. More importantly, healthy lifestyle behaviors at an early age have positive influences on reducing the risks of comorbidities in advanced years [6,7,8]. Some studies have shown that during the early adulthood period, adolescents and young adults tend to develop long-lasting health-related lifestyle behaviors [6,7,8].

The prevalence of negative lifestyle behaviors has been increasing globally [9,10,11,12,13,14]. However, there are no comprehensive reviews predicting the prevalence and direct effect of such behaviors among university students. The best evidence available comes from individual country-specific studies. In Turkey, a study found that lifestyle behaviors among university students need to improve [9]. Another recent study in Germany reported low physical activity levels and poor adherence to healthier dietary habits among university students [9,10]. In Saudi Arabia, numerous studies have indicated that young adults, especially university students, have high stress levels, unhealthy dietary practice, insufficient sleep, sedentary lifestyles, and low physical activity [11,12,13,14].

Usually, lifestyle behaviors are investigated as separate domains, thus losing important key health information about the clustering effect of these behaviors. Studies that examined the whole spectrum of lifestyle behaviors showed that, indeed, negative lifestyle behaviors tend to cluster [15,16,17]. This clustering pattern of different lifestyle behaviors could potentially affect the efficacy and effectiveness of lifestyle intervention, especially in university students [18]. These studies were conducted in populations with different cultures and moral values than those in the Middle East; therefore, their results cannot be generalized globally without further assessment [19]. In addition, the literature typically tends to focus on the association between lifestyle behavior and sociodemographic characteristics. For example, Princess Nourah bint Abdulrahman University (PNU) has conducted several nutritional awareness programs; they assessed the nutritional practice among university students and its association with dietary practice and sociodemographic characteristics [20]. Although the study reported unhealthy dietary practices among students, it did not assess other lifestyle variables [20]. Furthermore, only a limited number of studies have examined the complete cluster of lifestyle behaviors among young adults and related lifestyle behaviors with lifestyle-related disorders in Saudi Arabia [12,14,21,22]. These studies indicated that at least 30% to 85% of the students were physically inactive or sedentary, up to 95% did not get enough sleep, up to 70% were stressed, and 50% to 90% did not engage in healthy-eating habits.

There are numerous factors that contribute to pain incidence and severity, namely, physiological, physical, social, and psychological factors [23]. Furthermore, some studies have shown that pain presence and severity can alter lifestyle behaviors and vice versa [24,25,26]. However, the question of pain in relation to lifestyle behaviors in young adults remains open.

Undoubtedly, understanding the complete lifestyle behavior profile can help universities design effective in-campus interventions and assess the overall health impact on the risk of disease development [20]. Moreover, individual sociodemographic characteristics, such as age or educational level, the presence of comorbidities, and pain perception, can have a direct influence on the choice of lifestyle behaviors. Many studies have examined lifestyle habits in other populations, but few have focused on Saudi Arabia. To achieve practical outcomes for native populations, it is important to evaluate cultural and moral values, which significantly impact lifestyle choices. The relationship between youth behaviors and pain has received less research. Understanding the effects of lifestyle behaviors on pain presence and intensity, as well as the reciprocal effect of pain on lifestyle behaviors, may help develop pain management methods and improve well-being. Finally, factors affecting lifestyles among Saudi university students must be researched. This study seeks to fill knowledge gaps, offer context-specific perspectives, and advance treatments for youth holistic health and well-being. Recognizing and addressing the unique challenges of this demographic can reduce the risk of multiple health conditions and promote long-term healthy lifestyle choices. Therefore, this study aimed to evaluate self-reported lifestyle behaviors and their associations with sociodemographics, comorbidities, and pain among university students at PNU, utilizing a cross-sectional study design. PNU was chosen because it represents the largest women’s university in Saudi Arabia.

## 2. Methodology

### 2.1. Study Design

A cross-sectional study was chosen to better-assess the study objective among university students at Princess Nourah Bint Abdulrahman University (PNU) in Riyadh, Saudi Arabia. All the students who attended PNU during 2021 and 2022 academic year were considered eligible to participate in the study. No exclusion criteria were considered.

### 2.2. Participants and Recruitment

All enrolled students (approximately twenty-seven thousand students) at the beginning of the academic year were invited to complete an online questionnaire. The study invitation was sent to all the students via their university emails and pinned on their Blackboard (Learning Management System) homepage. Further, faculty members were asked to announce the study invitation at the end of lectures. No students were excluded from the data analysis. However, not all the students participated in the study, and only those who volunteered or gave their consent were included in the study.

### 2.3. Ethical Considerations and Consent

Ethical approval was obtained from the Institutional Review Board of PNU (IRB Log number 21-0375). Students completed the questionnaire using Google Forms from November 2021 to March 2022. Interested students accessed the study questionnaire electronically. Due to the nature of study assessment tool, a self-reported online questionnaire, informed consent was obtained electronically by clicking “I agree” to participate button at the beginning of the questionnaire. Students were given the choice of not participating by clicking “I don’t agree”, and no response was recorded. Students were informed electronically on the questionnaire welcome page about the purpose of the study and the confidentiality of their responses before informed consent was obtained.

### 2.4. Sample Size and Sampling Method

The sample size (543 students) was calculated using the following sample size equation for proportions: *n* = (z^2^pq)/d^2^). We assumed a population proportion that yields the maximum possible sample size required (*p* = 0.50), with a confidence level of 98% and a margin of error of 5%. An additional 15% of participants were added to account for the clustered design effect, nonresponders, and missing data. Thus, the effective sample size was finally set at 625 students. Convenience sampling technique was used to include the participants in this study.

### 2.5. Study Questionnaire

The study questionnaire consisted of four parts. The first part included general information questions such as age, marital status, educational level and major, citizenship, and social difficulties (caring for an infant or elderly person or facing social or financial difficulties). The second part of the questionnaire consisted of health-related information (i.e., selecting the presence or absence of a comorbidity from comprehensive comorbidities list, mental health symptoms (depression, anxiety, fears, shyness, and obsessions), and pain presence). The third part of the questionnaire covered the type of desired health promotion activities during the academic year, whereas the fourth part assessed lifestyle behaviors.

Lifestyle behaviors were assessed using Health-Promoting Lifestyle Profile II (HPLP-II), which was the last part of the questionnaire [27]. HPLP-II is a 4-point Likert scale with 52 items, including 6 subscales: health responsibility (9 items), spiritual growth (9 items), physical activity (8 items), interpersonal relationships (9 items), nutrition (9 items), and stress management (8 items). A Likert-type scale was used to measure each behavior, with ranges of never (1), sometimes (2), frequently (3), and regularly (4). The total score of the HPLP II ranges from 52 to 208 and is measured using the mean score of the responses to all 52 items. The authors use the mean of the subscale in addition to the total scores. A higher score in HPLP-II shows a higher level of health-promoting behavior. The total HPLP II score classifies lifestyle health as poor (52–90), moderate (91–129), good (130–168), and excellent (169–208). Further, 2 questions were added to assess sedentary behavior and sleep duration. HPLP II was used to assess lifestyle behaviors to help in determining which lifestyle-promoting behavior is associated with study variables to target in future university health promotion interventions.

All the questionnaire items were closed-looped questions except for age, sleep duration, and sedentary behavior duration. In these three questions, students were asked to enter their answers using numbers.

### 2.6. Data and Statistical Analyses

JMP Pro, version 14.2, was used to analyze the data. The sociodemographic characteristics, total HPLP II scores and subscales, sedentary behavior, and sleep duration were described using percentages, means, standard deviations, minimums, and maximums. Independent samples and analysis of variance (ANOVA) tests were applied where appropriate to compare the mean scores of total HPLP II, sedentary behavior, and sleep among marital status, social difficulties, mental health, sleep disorders, diseases, disability, pain, and genitourinary system symptoms. Pearson’s correlation coefficient analysis was used to assess relationships between ages, sedentary behavior, sleep duration, and HPLP II. Multiple linear regression test was used to study the quantitative effects of different independent variables, such as marital status, social difficulties, mental health, sleep disorders, diseases, disability, pain, and genitourinary system symptoms, on total HPLP II and subscales scores, sedentary behavior, and sleep duration. *p*-value was set as <0.05.

## 3. Results

A total of 1112 PNU students participated in the study. The sociodemographic factors are illustrated in Table 1, which also shows lifestyle behavior and students’ HPLP II scores for the six subscales. The age range was 17–43 years, with a mean of 19.8 ± 2.8 years. Nearly more than 3/4 of the participants (82.2%) were undergraduate students, 16.3% were diploma students, and only 1.5% were graduate students. The responders were enrolled in different colleges of PNU; the highest responders belonged to the College of Humanities (41.72%), followed by the College of Science (25.18%). Responders from the College of Health Science were below 20% (18.80%), while participants from Deanships and Institutes (9.17%) and the Applied College (5.13%) were less than 10%. Among the participants, 93.8% were single, and 4.7% were married, followed by 14 students who were divorced and 2 widows. More than half of the participants (56%) did not have a medical file in their neighborhood primary care center. As for social difficulties, 29.1% reported that they have social difficulties. Additionally, 54.7% of the participants stated that they had mental health issues, whereas a significant number of participants (30.7%) complained of having a sleep disorder.

In terms of disease occurrence, only 19.5% did not have any form of disease, whilst the most common reported diseases were those related to nutritional deficiencies (69%). This is followed by disorders of the digestive system and respiratory system (29.7% and 13.5%, respectively). The least reported diseases were cancer and renal and hereditary blood disorders (0.1%, 0.7%, and 1%, respectively). Moreover, only 2.8% reported some form of disability, including physical disability (0.5%) and auditory impairment (2.2%). Most of the participants (51.9%) also complained about pain symptoms, and about 27.3% reported symptoms related to the genitourinary system. The lifestyle behavior metrics of the responders revealed an average of 7.05 ± 2.46 h of sleep per day, and the average hours of sedentary behavior was 9.51 ± 4.93 per day. The HPLP II total mean score was 123.85 ± 62.41 (range =  52–208 score), and the highest mean in the subscales was 25.74 ± 6.44 for spiritual growth; this is closely followed by interpersonal relationships (24.87 ± 5.41). The mean of the subscales for nutrition, health responsibility, and stress management were 19.86 ± 5.23, 19.04 ± 5.78, and 18.95 ± 4.91, respectively. Interestingly, the lowest mean in the subscales of HPLP was for physical activity (15.35 ± 5.38).

It is evident from the HPLP II total, sleep, and sedentary behavior data presented in Table 2 that Deanships and Institutes, the College of Science, and the College of Humanities students showed worse sedentary behavior than the College of Health Science and the Applied College. In contrast, there was no significant difference between the colleges in the HPLP II total scores and sleep duration. The results also showed that divorced students and married students spent most of their time sitting compared to single and widowed students. Widowed students had the highest mean score of HPLP II (142 ± 2.83); however, there were no significant differences between all marital statuses. Additionally, the mean sleeping hours were similar between all marital categories; thus, no significant difference was found between the categories. In addition, the HPLP II total mean score of students reporting mental health issues was significantly lower than that of students without mental health problems (115.34 ± 23.58 vs. 134.16 ± 25.99; *p* < 0.0001). In contrast, the average sedentary hours among students with reported mental health problems was statistically significantly higher than in students reporting no mental health problems (9.86 ± 5.09 vs. 9.09 ± 4.69; *p* = 0.0093). Moreover, students who did not report sleep disorders have statistically significantly higher mean HPLP II scores compared with those with sleep disorders (128.6 ± 26.42 score vs. 113.13 ± 23.04 score; *p* < 0.0001). Also, sleeping hours for those who had no sleep disorders (7.33 ± 2.25 h) was significantly higher (*p* < 0.0001) compared with those with sleep disorders (6.42 ± 2.79 h). Students without any diseases had a significantly higher HPLP score (129.26 ± 26.64; *p* = 0.0017) than those who reported otherwise. With regards to disability, the average HPLP II total scores for physical disability (150.33 ± 21.15) were significantly (*p* = 0.0479) higher than those with auditory impairment (124.12 ± 27.47), as well as those with no disability (123.7 ± 26.36).

Based on the multiple linear regression results in Table 3, the students at the College of Science appear to have good lifestyle behaviors (an increase in HPLP II total scores of 3.69). In contrast, students with mental health issues tend to have poorer lifestyle behavior and sedentary behavior, as evidenced by the low HPLP II total scores (7.87) and more sitting hours (0.37), respectively. Students with “No sleep disorders” tend to have positive impacts with better lifestyle behavior (an HPLP II total score of 4.09) and sufficient sleeping hours (0.53).

Based on the multiple linear regression results in Table 4, students from the College of Science seem to have good interpersonal relationships, spiritual growth, and stress management (better scores of 0.67, 0.95, and 0.79, respectively). No disability tends to reduce health responsibility, physical activity, nutrition, and stress management scores (low scores of 1.98, 2.01, 1.91, and 1.48, respectively). Further, auditory disability tends to reduce health responsibility (a lower score of 2.48).

Mental health issues had a significant negative impact on all HPLP II subclasses: health responsibility, physical activity, nutrition, interpersonal relationships, spiritual growth, and stress management (decreased HPLP II subclass scores of 1.01, 0.90, 0.98, 1.39, 2.35, and 1.22, respectively). In contrast, no sleep disorders were found to have a significant positive impact on all HPLP II subscales (HPLP II subclass scores of 0.44, 0.44, 0.61, 0.52, 0.99, and 0.87, respectively). Longer sleep duration had a positive impact on the stress management subscale (score of 0.33). The presence of social difficulties was correlated with the worst stress management subscale score (a lower stress management score of 0.32). A longer duration of sedentary behavior was a predictor of the worst scores in the HPLP II subclasses (lower scores of 0.11, 0.17, 0.12, 0.07, and 0.07, respectively).

## 4. Discussion

This cross-sectional study looked at lifestyle choices made by PNU students in various colleges throughout the academic year 2021–2022 and their relationships with sociodemographic characteristics, comorbid conditions, and pain status. Overall, students demonstrated high levels of sedentary behavior, acceptable sleep length, and modest participation in activities that promote health. The only factors that were linked to lower levels of health-promoting activities were mental health, the presence of diseases, sleep difficulties, and the absence of impairment. These findings are important to help decision-makers in developing targeted lifestyle interventions.

With a mean HPLP II total score of 123.85 ± 62.41, this study shows that students displayed a moderate level of health-promoting activities [27]. The spiritual growth subscale had the greatest mean score, followed by interpersonal relationships, while the physical activity subscale had the lowest mean score. These values are consistent with the findings of earlier research carried out among Saudi university students [13,28] and in line with the scores among students in Turkey [9,29] and Hong Kong [30]. The high score in the spiritual growth subscale may be attributed to the cultural values in Arabian society, which emphasize the importance of spirituality and faith in daily life. The high score in the interpersonal relationships’ subscale can be attributed to strong family and social ties, which are valued in Eastern cultures, emphasizing the importance of maintaining positive relationships with family and friends.

On the other hand, the low score in the physical activity subscale was expected, as several studies of Saudi young adults and youth indicated high levels of physical inactivity [12,31,32]. Data for this study were collected after the COVID-19 pandemic; therefore, the observed low score of the physical activity subscale might have been exacerbated by the COVID-19 pandemic, as recent systematic reviews indicated that physical activity was low before the COVID-19 pandemic [33] and further decreased after it [34]. Low scores of physical activities at this young age are a cause for concern, as physical inactivity is a major risk factor for various chronic diseases, including cardiovascular disease, diabetes, and obesity [35,36].

The study results also revealed that sedentary behavior was prevalent among students, with a mean of 9.51 ± 4.93 h per day. These findings are like those previously reported by other studies in Saudi Arabia [37,38]. The high levels of sedentary behavior reported here might be due to the nature of students’ lives, where it is expected that students sit during class time and while studying outside their classrooms. Yet, it is an area of major health concern, as studies have shown that sedentary behavior is an independent risk factor for cardio-metabolic diseases and all-cause mortality [3,4]. Both high levels of physical inactivity and sedentary behaviors could be a result of Saudi societal norms that rely heavily on transportation via motor vehicles or environmental factors, such as high temperatures most months of the year in the study location and the low level of neighborhood walkability.

The average hours of sleep were 7.05 ± 2.46 per day, which is consistent with the recommended hours of sleep for adults (7–9 h per day) [39]. However, 30.7% of students reported having a sleep disorder, with average sleep hours of 6.42 ± 2.79. These results indicated an increase in average sleeping hours compared to previous studies conducted on other female university students in Saudi Arabia (average sleep duration of 5.5 ± 1.5) [40]. Another study reported that about 70% of students (both genders) had insufficient sleep duration [41]. These conflicting results warrant further investigation in future studies, as sleep duration and quality are crucial for health [39]. Further, our results accentuate the need for health promotion initiatives that address the underlying causes of university students’ irregular sleeping patterns.

In contrast, there was a significant (*p* < 0.0001) positive correlation between the absence of sleep disorders and HPLP II (4.09). However, despite having poor sedentary behavior patterns, these participants reported getting more sleep. Similar findings have been reported in studies exploring the relationship between sleep quality and health behaviors [42,43]. Cultural norms in the study location encourage a more active social life after sunset to avoid hot weather conditions. This could contribute to the observed results of poor sleep. However, to fully comprehend how sleep, sedentary behavior, and health-promoting behaviors interact, further investigation is required.

Furthermore, the study revealed that there is a significant association between mental health conditions and various subclasses of HPLP II (*p* < 0.05 in all cases). These subclasses encompassed health responsibility (1.01), physical activity (0.90), nutrition (0.98), interpersonal relationships (1.39), spiritual growth (2.35), and stress management (1.22). The findings align with prior studies that have demonstrated a correlation between mental well-being and lifestyle choices [44,45]. There is a correlation between mental health conditions and suboptimal lifestyle behaviors, as indicated by the lower scores of HPLP II and increased sedentary behavior in affected individuals. Though these variables are correlated, it is difficult to assert whether mental health conditions influence HPLP II or vice versa due to the limitations of the cross-sectional study.

The presence of disease had a detrimental impact on HPLP II (low scores of −1.18), suggesting poor health-promoting behaviors. Also, similar insignificant (*p* > 0.05) outcomes with sleep (0.11) and sedentary behavior (0.12) were noted, which might have led to a lower level of health-promoting activity. These outcomes are consistent with previous research that emphasized the influence of chronic illnesses on behaviors that promote good health [46,47]. It is important to acknowledge that the presence of disease and sleep disorders can potentially restrict individuals’ ability to participate in and prioritize activities that promote good health.

Scores in subclasses like health responsibility, physical activity, nutrition, and stress management (low scores of 1.98, 2.01, 1.91, and 1.48, respectively) were found to be lower when there was no disability [14,44]. Additionally, a lower score for health responsibility (2.48) was linked to auditory disability [48]. These results are in line with the previous literature showing how disability affects self-care practices and health behaviors. Individuals with disabilities may face unique challenges that hinder their ability to engage in certain health-promoting behaviors.

The findings of this study clearly demonstrate the need for specific health intervention measures and health-promoting programs that are directed toward enhancing the lifestyles of university students. Firstly, the results strongly suggest a need for interventions that promote physical activity and reduce sedentary behavior among university students. These interventions may include the provision of and access to sports facilities, organizing physical activity events, promoting the benefits of regular physical activity on health and well-being, and encouraging active classrooms whenever possible. Secondly, this study indicates that the sleeping habits of university students need to improve. Offering sleep-related counseling, encouraging healthy lifestyle behaviors, and educating students about the value of getting good quality sleep should be an integral part of wholesome sleep-promoting interventions. Lastly, this study also calls for the promotion of university students’ mental health and well-being. Interventions may include counseling services, stress management workshops, and spreading awareness of the advantages of good mental health on academic achievement and overall well-being.

As in any study, the current work has several limitations. First, the cross-sectional design of this study makes it difficult to interpret causal relationships and temporal variations in students’ lifestyle patterns. However, it is important to establish baseline data to identify areas where health promotion activities are needed. Second, as all the data gathered for this study were self-reported, it is likely that the responses reflect a bias toward socially acceptable responses, and the participants’ responses may not reflect reality. Thus, a large sample size was included to reduce such bias. Lastly, this study was conducted in a single university in Riyadh, Saudi Arabia, which may limit the generalizability of the findings. Future studies may need to utilize objective measures to assess lifestyle behaviors within multi-university settings with a cohort or longitudinal study design.

## 5. Conclusions

In conclusion, this cross-sectional study found that PNU students demonstrated moderate levels of health-promoting activities, with high scores in the spiritual growth and interpersonal relationships subscales and low scores in the physical activity subscale. Sedentary behavior was prevalent among students, and the average hours of sleep were within the recommended range but with a significant percentage reporting sleep disorders. Mental health conditions were associated with lower scores in various subclasses of health-promoting activities. The presence of diseases and sleep disorders had a negative impact on health-promoting behaviors, while the absence of sleep disorders was positively correlated with health-promoting activities. Disabilities were also found to affect self-care practices and health behaviors. These findings highlight the need for health promotion initiatives that address sedentary behavior, sleep disorders, mental health, and the unique challenges faced by individuals with disabilities in promoting their health.

## Figures and Tables

**Table 1 healthcare-12-00154-t001:** Distribution of students’ demographic characteristics (N = 1112).

Characteristic	Subcategory	Mean ± SD or N (%)	
Age (Mean ± SD)		19.8 (2.8)	
Educational Level	Diploma	181 (16.3)	
	Undergraduate	914 (82.2)	
	Graduate	17 (1.5)	
Colleges	College of Humanities	464 (41.72)	
	College of Science	280 (25.18)	
	College of Health Science	209 (18.80)	
	Deanships and Institutes *	102 (9.17)	
	Applied College	57 (5.13)	
Marital Status	Single	1043 (93.8)	
	Married	53 (4.7)	
	Divorced	14 (1.3)	
	Widowed	2 (0.2)	
With a File in Neighborhood Primary Care Center (yes)		489 (44.0)	
Social Difficulties	Yes	324 (29.1)	
Mental Health Symptoms	Yes	608 (54.7)	
Sleep Disorders	Yes	341 (30.7)	
Diseases	No diseases	217 (19.5)	
	Digestive system (yes)	330 (29.7)	
	Nervous system (yes)	35 (3.1)	
	Cardiovascular system (yes)	44 (4)	
	Respiratory system (yes)	150 (13.5)	
	Nutrition deficiencies and related conditions (yes)	767 (69)	
	Migraine (yes)	112 (10.1)	
	Renal disease (yes)	8 (0.7)	
	Hereditary blood disease (yes)	11 (1.0)	
	Diabetes (yes)	13 (1.2)	
	Cancer (yes)	1 (0.1)	
	Thyroid-related (yes)	24 (2.2)	
	Allergy (yes)	146 (13.1)	
Disability	No	1081 (97.2)	
	Physical	6 (0.5)	
	Auditory impairment	25 (2.2)	
Pain Symptoms	Yes	577 (51.9)	
Genitourinary System Symptoms	Yes	304 (27.3)	
Lifestyle Behaviors	Mean ± SD	(Min–Max)	
	Sleep (hours)	7.05 ± 2.46	(1–24)
	Sedentary behavior (hours)	9.51 ± 4.93	(1–24)
HPLP II and Subscales	Mean ± SD	(Min–Max)	
	Health responsibility (9)	19.04 ± 5.78	(9–36)
	Physical activity (8)	15.35 ± 5.38	(8–32)
	Nutrition (9)	19.86 ± 5.23	(9–36)
	Spiritual growth (9)	25.74 ± 6.44	(9–36)
	Interpersonal relationships (9)	24.87 ± 5.41	(9–36)
	Stress management (8)	18.95 ± 4.91	(8–32)
	Total HPLP II (52)	123.85 ± 62.41	(52–208)

Social difficulties: caring for an infant or elderly person or facing social or financial difficulties. Mental health symptoms: depression, anxiety, fears, shyness, and obsessions. Genitourinary system symptoms: heavy menstrual flow, irregular menstrual cycle, or painful mensural cramps. HPLP II: Health-Promoting Lifestyle Profile II * refers to the group of responders from different deanships and institutes.

**Table 2 healthcare-12-00154-t002:** Distribution of Health-Promoting Lifestyle Profile (HPLP) scores, sleep, and sedentary behavior according to colleges, marital status, comorbidities, and pain symptoms (N = 1112).

	HPLP II Total		Sleep		Sedentary Behavior	
	Mean ± SD	*p*-Value	Mean ± SD	*p*-Value	Mean ± SD	*p*-Value
Colleges						
College of Humanities	124.10 ± 26.71	0.264	7.08 ± 2.46	0.346	9.58 ± 4.78	0.0464 *
College of Science	125.39 ± 26.80		7.01 ± 2.20		9.65 ± 4.99	
College of Health Science	122.68 ± 24.28		7.15 ± 2.90		8.96 ± 4.92	
Deanships and Institutes	119.23 ± 26.15		6.65 ± 2.07		10.55 ± 5.49	
Applied College	126.79 ± 29.58		7.42 ± 2.64		8.54 ± 4.69	
Marital Status						
Single	123.81 ± 26.25	0.637	7.04 ± 2.46	0.662	9.41 ± 4.85	0.0107 *
Married	122.55 ± 28.75		7.08 ± 2.79		10.81 ± 5.93	
Divorced	129.43 ± 31.35		7.86 ± 1.75		13 ± 5.46	
Widowed	142 ± 2.83		7.50 ± 0.71		9 ± 1.41	
Comorbidities						
Social Difficulties						
No	123.63 ± 26.48	0.676	6.99 ± 2.37	0.262	9.36 ± 4.82	0.106
Yes	124.37 ± 26.28		7.19 ± 2.68		9.9 ± 5.18	
Mental Health						
No	134.16 ± 25.99	<0.0001 *	7.12 ± 2.15	0.3702	9.09 ± 4.69	0.0093 *
Yes	115.34 ± 23.58		6.99 ± 2.69		9.86 ± 5.09	
Sleep Disorders						
No	128.6 ± 26.42	<0.0001 *	7.33 ± 2.25	<0.0001 *	9.38 ±4.87	0.1633
Yes	113.13 ± 23.04		6.42 ± 2.79		9.83 ±5.06	
Diseases						
Absence of Diseases	129.26 ± 26.64	0.0017 *	6.9 ± 2.07	0.5523	9.47 ± 4.98	0.9425
One Disease	124.11 ± 27.25		7.13 ± 2.55		9.61 ± 4.83	
Two Diseases or More	121.85 ± 25.72		7.07 ± 2.55		9.49 ± 4.97	
Disability						
No	123.7 ± 26.36	0.0479 *	7.05 ± 2.46	0.5295	9.52 ± 4.95	0.6916
Physical	150.33 ± 21.15		6.17 ± 1.6		11 ± 4.56	
Auditory Impairment	124.12 ± 27.47		7.4 ± 2.7		9.08 ± 4.42	
Symptoms						
Pain						
No	124.75 ± 27	0.2749	7.03 ± 2.42	0.7440	9.34 ± 4.85	0.2464
Yes	123. 01 ± 25.85		7.08 ± 2.51		9.68 ± 5	
Genitourinary System						
No	124.29 ± 26.95	0.3456	6.97 ± 2.32	0.0666	9.54 ± 4.91	0.7610
Yes	122.67 ± 24.92		7.29 ± 2.81		9.44 ± 4.99	

* Statistically significant at *p*-value < 0.05. Social difficulties: caring for an infant or elderly person or facing social or financial difficulties. Mental health symptoms: depression, anxiety, fears, shyness, and obsessions. Genitourinary system symptoms: heavy menstrual flow, irregular menstrual cycle, or painful mensural cramps. HPLP II: Health-Promoting Lifestyle Profile II.

**Table 3 healthcare-12-00154-t003:** Multiple linear regression showing the possible predictors of Health-Promoting Lifestyle Profile scores, sleep hours, and sedentary behavior in students.

Factors	HPLP II Total					Sleep					Sedentary Behavior				
	Estimate (β)	SE	*p*-Value	95% CI		Estimate (β)	SE	*p*-Value	95% CI		Estimate (β)	SE	*p*-Value	95% CI	
				Lower	Upper				Lower	Upper				Lower	Upper
Intercept	137.70	6.04	<0.0001 *	125.85	149.55	7.12	0.60	<0.0001 *	5.94	8.30	10.73	1.21	<0.0001 *	8.35	13.11
Colleges															
Applied College	1.25	2.69	0.6416	−4.03	6.54	0.29	0.27	0.2773	−0.23	0.82	−0.81	0.54	0.1352	−1.87	0.25
College of Health Science	−1.37	1.64	0.4014	−4.58	1.II84	0.13	0.16	0.438	−0.19	0.45	−0.41	0.33	0.2093	−1.06	0.23
College of Humanities	0.60	1.30	0.6477	−1.96	3.15	0.03	0.13	0.7991	−0.22	0.29	0.15	0.26	0.5755	−0.37	0.66
College of Science	3.69	1.49	0.0133 *	0.77	6.61	−0.02	0.15	0.9147	−0.31	0.27	0.18	0.30	0.5426	−0.40	0.77
Marital Status															
Divorced	−1.30	6.65	0.8455	−14.34	11.75	0.32	0.66	0.6291	−0.98	1.62	2.45	1.34	0.0669	−0.17	5.07
Married	−5.78	5.29	0.2749	−16.17	4.60	−0.42	0.53	0.4301	−1.45	0.62	0.06	1.06	0.9582	−2.03	2.14
Single	−5.38	4.77	0.2601	−14.74	3.99	−0.43	0.47	0.3624	−1.36	0.50	−1.02	0.96	0.2895	−2.90	0.87
Diseases															
No Diseases	1.93	1.32	0.1453	−0.67	4.52	−0.17	0.13	0.197	−0.43	0.09	0.17	0.27	0.5121	−0.35	0.70
One Disease	−1.18	1.18	0.3197	−3.50	1.14	0.11	0.12	0.3393	−0.12	0.34	0.12	0.24	0.6201	−0.35	0.58
Disability															
Auditory Impairment	−3.75	4.70	0.4254	−12.98	5.48	0.56	0.47	0.23	−0.36	1.48	−1.03	0.95	0.2756	−2.89	0.82
No Disability	−9.45	3.75	0.0118 *	−16.80	−2.10	0.21	0.37	0.5778	−0.52	0.94	−0.27	0.75	0.7238	−1.74	1.21
Symptoms															
No Genitourinary System Symptoms	0.03	0.88	0.9695	−1.69	1.76	−0.17	0.09	0.0573	−0.34	0.01	0.17	0.18	0.3413	−0.18	0.51
No Pain Symptom	−0.18	0.80	0.8222	−1.75	1.39	0.02	0.08	0.824	−0.14	0.17	−0.29	0.16	0.0761	−0.60	0.03
Mental Health															
Mental Health	−7.87	0.83	<0.0001 *	−9.49	−6.25	0.12	0.08	0.1602	−0.05	0.28	0.37	0.17	0.027 *	0.04	0.69
Sleep Disorders															
No Sleep Disorders	4.09	0.88	<0.0001 *	2.36	5.83	0.53	0.09	<0.0001 *	0.35	0.70	−0.08	0.18	0.6627	−0.43	0.27
Social Difficulties															
No Social Difficulties	−1.08	0.86	0.2096	−2.77	0.61	-0.07	0.09	0.4072	−0.24	0.10	−0.18	0.17	0.3116	−0.51	0.16
F-value	12.976					3.218					1.941				
*p*-value	<0.0001 *					<0.0001 *					0.0074 *				
R Square	0.160					0.045					0.028				

* Statistically significant at *p*-value < 0.005. Social difficulties: caring for an infant or elderly person or facing social or financial difficulties. Mental health symptoms: depression, anxiety, fears, shyness, and obsessions. Genitourinary system symptoms: heavy menstrual flow, irregular menstrual cycle, or painful mensural cramps. HPLP II: Health-Promoting Lifestyle Profile II.

**Table 4 healthcare-12-00154-t004:** Multiple Linear Regression showing predictors of Health-Promoting Lifestyle Profile scores in Students.

Factors	Health Responsibility					Physical Activity					Nutrition					Interpersonal Relationships					Spiritual Growth					Stress Management				
	Estimate (β)	SE	*p*-Value	95% CI		Estimate (β)	SE	*p*-Value	95% CI		Estimate (β)	SE	*p*-Value	95% CI		Estimate (β)	SE	*p*-Value	95% CI		Estimate (β)	SE	*p*-Value	95% CI		Estimate (β)	SE	*p*-Value	95% CI	
				Lower	Upper				Lower	Upper				Lower	Upper				Lower	Upper				Lower	Upper				Lower	Upper
Intercept	21.56	1.05	<0.0001 *	19.50	23.62	19.08	0.97	<0.0001 *	17.17	20.98	22.64	0.94	<0.0001 *	20.80	24.49	25.64	0.96	<0.0001 *	23.75	27.53	27.03	1.08	<0.0001 *	24.92	29.15	18.61	0.84	<0.0001 *	16.97	20.25
collages																														
Applied collage	0.06	0.62	0.9275	−1.16	1.27	0.53	0.57	0.3564	−0.59	1.65	0.23	0.55	0.6728	−0.85	1.32	0.14	0.57	0.8032	−0.97	1.25	−0.53	0.64	0.4087	−1.77	0.72	0.28	0.49	0.5669	−0.68	1.25
Colleges of health science	−0.48	0.37	0.1959	−1.22	0.25	−0.33	0.35	0.3411	−1.01	0.35	0.02	0.34	0.9538	−0.64	0.68	−0.12	0.34	0.7184	−0.80	0.55	−0.28	0.39	0.472	−1.03	0.48	−0.57	0.30	0.0553	−1.16	0.01
Colleges of Humanities	0.19	0.30	0.5182	−0.39	0.78	0.16	0.28	0.564	−0.38	0.70	0.12	0.27	0.6663	−0.41	0.64	−0.07	0.27	0.8108	−0.60	0.47	0.19	0.31	0.5327	−0.41	0.80	0.11	0.24	0.6535	−0.36	0.57
Colleges of Sciences	0.61	0.34	0.0757	−0.06	1.28	0.29	0.32	0.3524	−0.33	0.91	0.35	0.31	0.255	−0.25	0.95	0.67	0.31	0.0316 *	0.06	1.29	0.95	0.35	0.0069 *	0.26	1.64	0.79	0.27	0.0038 *	0.26	1.32
Disability																														
Auditory impairment	−2.48	1.08	0.0216 *	−4.59	−0.36	−0.46	1.00	0.6422	−2.42	1.49	0.31	0.97	0.7498	−1.59	2.20	−1.23	0.99	0.2154	−3.17	0.72	−0.48	1.11	0.6646	−2.66	1.70	−0.22	0.86	0.7939	−1.91	1.46
No Disability	−1.98	0.86	0.0213 *	−3.67	−0.30	−2.01	0.80	0.0118 *	−3.57	−0.45	−1.91	0.77	0.0134 *	−3.42	−0.40	−0.63	0.79	0.4252	−2.18	0.92	−1.47	0.88	0.0971	−3.20	0.27	−1.48	0.68	0.0303 *	−2.83	−0.14
Mental health																														
Mental health	−1.01	0.19	<0.0001 *	−1.38	−0.64	−0.90	0.17	<0.0001 *	−1.24	−0.55	−0.98	0.17	<0.0001 *	−1.31	−0.65	−1.39	0.17	<0.0001 *	−1.73	−1.05	−2.35	0.19	<0.0001 *	−2.73	−1.97	−1.22	0.15	<0.0001 *	−1.51	−0.92
Sleep disorders																														
No sleep disorders	0.44	0.21	0.0306 *	0.04	0.85	0.44	0.19	0.0213 *	0.07	0.81	0.61	0.18	0.0009 *	0.25	0.97	0.52	0.19	0.0058 *	0.15	0.89	0.99	0.21	<0.0001 *	0.58	1.41	0.87	0.16	<0.0001 *	0.55	1.19
Social difficulties																														
No Social difficulties	−0.09	0.19	0.6453	−0.46	0.28	−0.12	0.17	0.4744	−0.47	0.22	−0.14	0.17	0.4038	−0.47	0.19	−0.17	0.17	0.3381	−0.51	0.17	−0.22	0.19	0.2562	−0.60	0.16	−0.32	0.15	0.032 *	−0.62	−0.03
Sleep	0.05	0.07	0.4739	−0.09	0.19	−0.03	0.06	0.6373	−0.16	0.10	0.01	0.06	0.8535	−0.11	0.13	0.05	0.06	0.4225	−0.07	0.18	0.08	0.07	0.2866	−0.06	0.22	0.33	0.06	<0.0001 *	0.22	0.44
Sedentary Behavior	−0.11	0.03	0.0013 *	−0.18	−0.04	−0.17	0.03	<0.0001 *	−0.24	−0.11	−0.12	0.03	<0.0001 *	−0.19	−0.06	−0.06	0.03	0.0646	−0.12	0.00	−0.07	0.04	0.0486 *	−0.14	0.00	−0.07	0.03	0.0066 *	−0.13	−0.02
F-value	7.446					8.750					9.742					11.679					25.771					22.064				
*p*-value	<0.0001 *					<0.0001 *					<0.0001 *					<0.0001 *					<0.0001 *					<0.0001 *				
R Square	0.069					0.081					0.089					0.105					0.205					0.181				

* Statistically significant at *p* value < 0.005.

## Data Availability

The datasets generated and/or analyzed during the current study are available from the corresponding author on reasonable request.

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
