# Peer review of "Factors Associated with Lifestyle Behaviors among University Students—A Cross-Sectional Study"

_healthcare, 2024, doi:10.3390/healthcare12020154_

Round 1
Reviewer 1 Report
Comments and Suggestions for Authors
The introduction is clear and concise. The authors provide a background on relevant studies and discuss the study’s important findings and limitations. The authors also propose future recommendations.
However, it is suggested to review the following aspects:
(i) the abstract should be simplified and the mention to tables removed;
(ii) In line 167, the percentage of auditory impairment (2.2%) should be corrected, and in lines 176 and 177, the values presented are distinct from the ones shown in Table 2. According to the data in this table, there is no correlation between student´s age and HPLP II and subscales (this data also needs to be reviewed in the abstract).
Author Response
Reply to the comments of Reviewer 1
Comment: The introduction is clear and concise. The authors provide a background on relevant studies and discuss the study’s important findings and limitations. The authors also propose future recommendations.
Reply: The authors are grateful for the compliments and appreciation from the respected Reviewer 1.
Comment: However, it is suggested to review the following aspects: (i) the abstract should be simplified and the mention to tables removed.
Reply: The suggestions from the reviewer 1 have been accepted and the authors have made the changes.
Comment: (ii) In line 167, the percentage of auditory impairment (2.2%) should be corrected, and in lines 176 and 177, the values presented are distinct from the ones shown in Table 2.
Reply: The suggested correction has been made in line 167. Additionally, we would like to clarify that the data mentioned in lines 176 and 177 are referred from Table 1 and not Table 2.
Comment: According to the data in this table, there is no correlation between student´s age and HPLP II and subscales (this data also needs to be reviewed in the abstract).
Reply: The suggestion has been accepted by the authors and the authors appreciate the feedback from the knowledgeable Reviewer 1.
Reviewer 2 Report
Comments and Suggestions for Authors
Dear Authors,
I appreciate your submission of the manuscript “ Factors Associated with Lifestyle Behaviours Among University Students –A Cross-Sectional Study ”. Your research addresses an important topic in the field and contributes valuable insights.
After thoroughly reviewing your manuscript, I offer suggestions for improving your work’s quality, clarity, and impact. Please remember that these are recommendations, and you can decide how best to address them based on your understanding of the subject matter and your intended audience. Here are the suggestions:
abstract
- Clarity of Purpose and Scope:
- The abstract should clearly state the study’s primary objective or research question. What specifically were the researchers trying to investigate? In this case, it's about lifestyle behaviors among university students, but it would be helpful to be more explicit about the main aim.
- Methodology:
- The abstract briefly mentions using Pearson's correlation and multiple linear regression, but it should provide more detail about the methods used in the study. Saying the sample size, data collection methods, and any significant statistical tests or measures would be beneficial for readers to understand how the study was conducted.
- Key Findings:
- While the abstract mentions some correlations and results, it could be more explicit about the key findings. For example, instead of just saying positive or negative correlations, briefly summarize the main findings related to lifestyle behaviours and their associations with socio-demographics, comorbidities, and pain. Use specific values or percentages to provide a clearer picture of the results.
- Significance and Implications:
- The abstract should include a sentence or two discussing the significance of the study and its potential implications. Why are these findings significant, and how might they be used to benefit students or the university community? What are the broader implications for health promotion interventions?
- Limitations:
- It's essential to acknowledge any limitations of the study in the abstract. Were there any limitations in the data, methodology, or sample that could affect the validity of the findings? Being transparent about limitations adds credibility to the research.
- Conclusion:
- Conclude the abstract by summarizing the main takeaway or message of the study. What should readers remember after reading the abstract? In this case, it could be a clear statement about the need for health promotion interventions among Saudi university students based on the study's findings.
- Formatting:
- Ensure that the abstract is well-structured and follows a logical sequence. Use clear headings or subheadings if necessary to organize the content.
Introduction
- Clearly Define the Research Problem:
- The introduction should start by clearly defining the research problem or gap in the existing literature. What specific aspect of lifestyle behaviours among university students is the study addressing? Is there a particular research question or hypothesis the study aims to answer? Providing a clear research focus will help readers understand the significance of the study.
- Provide a Rationale for the Study:
- Explain why investigating lifestyle behaviours among university students is essential. What are the potential health implications, especially for young adults? Is there any particular reason why Princess Nourah Bint Abdulrahman University (PNU) students were chosen for this study?
- Review of Relevant Literature:
- While the introduction mentions some studies from Turkey, Germany, and Saudi Arabia, providing a more comprehensive review of relevant literature on lifestyle behaviours among university students would be helpful. This could include statistics, trends, and key findings from previous research to establish a stronger foundation for the study.
- Justify the Need for a Holistic Approach:
- The introduction mentions that lifestyle behaviours have been investigated as separate domains. Still, it must provide a clear rationale for why a holistic approach, considering multiple lifestyle factors together, is essential. Explain the potential benefits of understanding the clustering effect of these behaviours and how it can contribute to a more comprehensive assessment of students' health.
- Highlight the Study's Objectives:
- Clearly state the specific objectives of the study in the introduction. What are the researchers trying to achieve by examining lifestyle behaviours, comorbidities, and pain among university students at PNU? This will give readers a sense of what to expect from the study.
- Transition to the Methodology:
- The introduction could end with a smooth transition to the methodology section, briefly mentioning how the study was conducted, the data collection methods used, and the population under investigation.
Methodology
- Study Design:
- While the section mentions that a cross-sectional study was conducted at Princess Nourah Bint Abdulrahman University (PNU), it would be helpful to briefly explain why a cross-sectional design was chosen for this particular study. Mention the advantages of this design for studying lifestyle behaviours among university students.
- Participants and Recruitment:
- Provide information about the total number of students at PNU during the 2021 and 2022 academic years. This will help readers understand the population size from which the sample was drawn.
- Ethical Considerations and Consent:
- Mention the specific ethical considerations that were taken into account during the study. For example, did the researchers consider informed consent, privacy, and confidentiality issues? Explain how these ethical considerations were addressed in the study design.
- Sample Size Calculation:
- Explain the rationale behind choosing a 98% confidence level and a 5% margin of error for sample size calculation. Why were these specific values determined, and how do they relate to the research objectives? Also, please explain the clustered design effect and why it was considered in the sample size calculation.
- Study Questionnaire:
- Provide more details about the content of the study questionnaire. What questions were included in each part of the questionnaire related to age, comorbidities, pain, health promotion activities, and lifestyle behaviours? This will give readers a better understanding of the data collected.
- Health-Promoting Lifestyle Profile II (HPLP-II):
- Explain why the HPLP-II instrument was chosen for assessing lifestyle behaviours. Briefly describe the six sub-scales and how they contribute to determining health-promoting behaviour. Additionally, clarify why the mean of the sub-scales was used rather than the total scores.
- Data and Statistical Analyses:
- While the section mentions the statistical software used (JMP Pro, version 14.2), providing a brief rationale for why this software was chosen for data analysis would be beneficial. Additionally, consider providing some information on the specific statistical tests used, especially for the independent samples, analysis of variance (ANOVA), Pearson's correlation coefficient analysis, and multiple linear regression. This will help readers understand how the data were analyzed.
Results:
- Explanatory Notes: It would be helpful to include some explanatory notes or footnotes for the abbreviations and terms used in the table, such as "HPLP II" (Health-Promoting Lifestyle Profile II), to ensure that readers understand the meaning of these terms.
- Interpretation of Results: While the table presents a lot of numerical data, it's essential to provide some understanding of the key findings. For example, what do the correlations in Table 2 imply? How do they relate to the study's objectives or hypotheses? Providing some context and interpretation can help readers understand the significance of the results.
- Discussion of Disease Categories: In the section discussing disease categories, you may want to elaborate on why certain diseases are more prevalent among the student population. Are there any potential factors or trends that could explain these findings?
- Discussion of Subscale Scores: In the section where you present the subscale scores of the Health-Promoting Lifestyle Profile (HPLP II), you could provide some context or discussion about why specific subscales have higher or lower scores. Are there any implications for the student’s overall well-being or health behaviours?
- Multiple Linear Regression Interpretation: When presenting the results of multiple linear regression in Tables 4 and 5, provide interpretations of the coefficients. Explain what a positive or negative coefficient means in the context of the study and how it relates to the predictor variables.
- Statistical Significance: Indicate which results are statistically significant (e.g., using asterisks or symbols) to highlight the most important findings.
- Conclusion of the Results Section: Summarize the main findings of the results section before moving on to the discussion. This will help readers grasp the key takeaways from this section.
Discussion
Introduction and Context Setting:
- Start the discussion section with a brief recap of the study's main objectives and key findings to provide context for readers who may have not read the entire article.
- Mention the significance of the study and its potential implications for public health or university policies.
- Structure and Flow:
- Organize the discussion logically by discussing each factor or finding one at a time and then drawing connections between them.
- Use subheadings to delineate different aspects of the discussion, making it easier for readers to follow your arguments.
- Interpretation of Findings:
- Provide more in-depth interpretation and analysis of the study's findings. For example, discuss why certain factors, such as mental health or disease presence, are associated with lower health-promoting behaviours. What underlying mechanisms might explain these associations?
- Consider discussing the potential reasons behind the variation in lifestyle behaviours among different subgroups of students (e.g., cultural factors, societal norms, or campus environments).
- Comparison with Prior Research:
- Discuss how your study's findings compare and contrast with previous research in this area, both within and outside Saudi Arabia.
- Mention any trends or changes observed compared to studies conducted before and after the COVID-19 pandemic, as it may have influenced lifestyle behaviours.
- Limitations and Future Directions:
- Acknowledge the limitations of your study, such as potential biases or limitations in data collection methods. This demonstrates transparency.
- Suggest directions for future research. What questions remain unanswered, and how could future studies address them?
- Practical Implications:
- Discuss the practical implications of your findings. What can universities or policymakers do to promote healthier lifestyles among students, especially regarding observed sedentary behaviour and sleep disorders?
- Consider addressing potential interventions or strategies that could improve the health-promoting activities of university students.
- Conclusion:
- Summarize the key takeaways from your discussion, emphasizing the most important findings and their implications.
- Avoid introducing new information or data in the conclusion; it should primarily be a summary.
- Citations:
- Ensure you appropriately cite previous research when comparing or drawing on prior findings.
Recommendations:
- Specificity and Clarity: While you provide some valuable recommendations, consider making them more specific and actionable. For example, instead of suggesting "promoting physical activity," specify the types of activities or programs that could be introduced, such as yoga classes, fitness challenges, or intramural sports.
- Prioritization: Please prioritize your recommendations based on the findings. For instance, if sedentary behaviour is a more significant concern than sleep patterns, emphasize interventions to reduce sedentary behaviour first.
- Evidence-Based Interventions: Support your recommendations with evidence from existing research or best practices in promoting health and well-being among university students. This adds credibility to your suggestions.
- Inclusivity: Consider addressing the diverse needs of university students. Not all students have the same lifestyle patterns or require the same interventions. Mention strategies for engaging various subgroups, such as international students, students with disabilities, or those from different cultural backgrounds.
- Feasibility: Discuss the feasibility and potential challenges of implementing these recommendations in a university setting. This can help institutions assess the practicality of adopting such interventions.
Limitations:
- Causal Inference: In the "Limitations" section, you correctly point out the limitations of the cross-sectional design. However, you can also suggest how future research, such as longitudinal studies, could address these limitations and provide insights into causal relationships.
- Socially Acceptable Responses: Acknowledge that self-reported data can be subject to biases, but also mention how the study attempted to mitigate this bias (e.g., by including large sample size) and suggest potential ways to improve data collection in future studies, such as using objective measurements or combining self-reporting with other data sources.
- Generalizability: While you mention that the study was conducted in a single university in Riyadh, Saudi Arabia, you can also discuss the potential cultural or regional factors that might influence the findings. This can help readers understand the context better and assess whether the results could apply to other settings.
- Ethical Considerations: Briefly mention any ethical considerations or limitations related to the study, such as informed consent, privacy, or potential participant biases.
Conclusion
- Summarize Key Findings: In this section, it's crucial to provide a concise summary of the main findings of your study. Mention the critical lifestyle behaviours identified as areas of concern among university students. This will help readers quickly grasp the most important takeaways from your research.
- Link to Research Questions/Hypotheses: If your study had specific research questions or hypotheses, briefly reiterate them and then state whether they were supported by your findings. This provides a clear link between your research objectives and the results.
- Practical Implications: Discuss the practical implications of your findings. How can the insights gained from this study be applied in real-world scenarios? How can policymakers and university administrators use this information to develop effective interventions?
- Recommendations for Future Research: Expand on the need for future research mentioned in the conclusion. Provide more specific suggestions for future studies, particularly those involving longitudinal analysis, to assess the effectiveness of interventions.
- Generalizability: Discuss the generalizability of your findings beyond the specific university in Saudi Arabia where the study was conducted. Mention any factors or limitations that may affect the applicability of your results to other settings or populations.
- Concluding Statement: End the conclusion with a strong concluding statement that highlights the broader significance of your study. Explain why addressing lifestyle behaviours among university students is essential, both from a public health perspective and for the student’s well-being.
- Avoid Repetition: The phrase "insinuate that there is a need for health promotion interventions" could be replaced with a more direct statement. Additionally, phrases such as "Further, the results of this study will undoubtedly assist policymakers and university administrators" could be made more concise and to the point.
Comments on the Quality of English Language
Overall, the English in the article is generally good, with only minor areas for improvement in terms of providing more context for the results and enhancing the recommendations and conclusion for greater clarity and impact.
Author Response
Reply to the comments of Reviewer 2
Comment: Dear Authors, I appreciate your submission of the manuscript “ Factors Associated with Lifestyle Behaviours Among University Students – A Cross-Sectional Study ”. Your research addresses an important topic in the field and contributes valuable insights. After thoroughly reviewing your manuscript, I offer suggestions for improving your work’s quality, clarity, and impact. Please remember that these are recommendations, and you can decide how best to address them based on your understanding of the subject matter and your intended audience.
Reply: The authors are pleased to receive the compliments and suggestions from the reviewer. The guidance of the reviewer has enhanced the quality of the presented manuscript.
Comment: Here are the suggestions: abstract Clarity of Purpose and Scope: The abstract should clearly state the study’s primary objective or research question. What specifically were the researchers trying to investigate? In this case, it's about lifestyle behaviors among university students, but it would be helpful to be more explicit about the main aim. Methodology: The abstract briefly mentions using Pearson's correlation and multiple linear regression, but it should provide more detail about the methods used in the study. Saying the sample size, data collection methods, and any significant statistical tests or measures would be beneficial for readers to understand how the study was conducted. Key Findings: While the abstract mentions some correlations and results, it could be more explicit about the key findings. For example, instead of just saying positive or negative correlations, briefly summarize the main findings related to lifestyle behaviours and their associations with socio-demographics, comorbidities, and pain. Use specific values or percentages to provide a clearer picture of the results. Significance and Implications: The abstract should include a sentence or two discussing the significance of the study and its potential implications. Why are these findings significant, and how might they be used to benefit students or the university community? What are the broader implications for health promotion interventions? Limitations: It's essential to acknowledge any limitations of the study in the abstract. Were there any limitations in the data, methodology, or   sample that could affect the validity of the findings? Being transparent about limitations adds credibility to the research. Conclusion: Conclude the abstract by summarizing the main takeaway or message of the study. What should readers remember after reading the abstract? In this case, it could be a clear statement about the need for health promotion interventions among Saudi university students based on the study's findings.
Reply: The suggestions and recommendations of the reviewer has been incorporated in the revised version of the manuscript.
Comment: Formatting: Ensure that the abstract is well-structured and follows a logical sequence. Use clear headings or subheadings if necessary to organize the content.
Reply: The suggestions and recommendations of the reviewer has been incorporated in the revised version of the manuscript.
Comment: Introduction Clearly Define the Research Problem: The introduction should start by clearly defining the research problem or gap in the existing literature. What specific aspect of lifestyle behaviours among university students is the study addressing? Is there a particular research question or hypothesis the study aims to answer? Providing a clear research focus will help readers understand the significance of the study. Provide a Rationale for the Study: Explain why investigating lifestyle behaviours among university students is essential. What are the potential health implications, especially for young adults? Is there any particular reason why Princess Nourah Bint Abdulrahman University (PNU) students were chosen for this study?
Reply: The suggestions and recommendations of the reviewer has been incorporated in the revised version of the manuscript.
Comment: Review of Relevant Literature: While the introduction mentions some studies from Turkey, Germany, and Saudi Arabia, providing a more comprehensive review of relevant literature on lifestyle behaviours among university students would be helpful. This could include statistics, trends, and key findings from previous research to establish a stronger foundation for the study. Justify the Need for a Holistic Approach: The introduction mentions that lifestyle behaviours have been investigated as separate domains. Still, it must provide a clear rationale for why a holistic approach, considering multiple lifestyle factors together, is essential. Explain the potential benefits of understanding the clustering effect of these behaviours and how it can contribute to a more comprehensive assessment of students' health. Highlight the Study's
Reply: The suggestions and recommendations of the reviewer has been incorporated in the revised version of the manuscript.
Comment: Objectives: Clearly state the specific objectives of the study in the introduction. What are the researchers trying to achieve by examining lifestyle behaviours, comorbidities, and pain among university students at PNU? This will give readers a sense of what to expect from the study. Transition to the Methodology:   The introduction could end with a smooth transition to the methodology section, briefly mentioning how the study was conducted, the data collection methods used, and the population under investigation.
Reply: The suggestions and recommendations of the reviewer has been incorporated in the revised version of the manuscript.
Comment: Methodology Study Design: While the section mentions that a cross-sectional study was conducted at Princess Nourah Bint Abdulrahman University (PNU), it would be helpful to briefly explain why a cross-sectional design was chosen for this particular study. Mention the advantages of this design for studying lifestyle behaviours among university students. Participants and Recruitment: Provide information about the total number of students at PNU during the 2021 and 2022 academic years. This will help readers understand the population size from which the sample was drawn.
Reply: The suggestions and recommendations of the reviewer has been incorporated in the revised version of the manuscript.
Comment: Ethical Considerations and Consent: Mention the specific ethical considerations that were taken into account during the study. For example, did the researchers consider informed consent, privacy, and confidentiality issues? Explain how these ethical considerations were addressed in the study design.
Reply: The suggestions and recommendations of the reviewer has been incorporated in the revised version of the manuscript.
Comment: Sample Size Calculation: Explain the rationale behind choosing a 98% confidence level and a 5% margin of error for sample size calculation. Why were these specific values determined, and how do they relate to the research objectives? Also, please explain the clustered design effect and why it was considered in the sample size calculation.
Reply: The suggestions and recommendations of the reviewer has been incorporated in the revised version of the manuscript.
Comment: Study Questionnaire: Provide more details about the content of the study questionnaire. What questions were included in each part of the questionnaire related to age, comorbidities, pain, health promotion activities, and lifestyle behaviours? This will give readers a better understanding of the data collected. Health-Promoting Lifestyle Profile II (HPLP-II): Explain why the HPLP-II instrument was chosen for assessing lifestyle behaviours. Briefly describe the six sub-scales and how they contribute to determining health-promoting behaviour. Additionally, clarify why the mean of the sub-scales was used rather than the total scores. Data and Statistical Analyses: While the section mentions the statistical software used (JMP Pro, version 14.2), providing a brief rationale for why this software was chosen for data analysis would be beneficial. Additionally, consider providing some information on the specific statistical tests used, especially for the independent samples, analysis of variance (ANOVA), Pearson's correlation coefficient analysis, and multiple linear regression. This will help readers understand how the data were analyzed.
Reply: The suggestions and recommendations of the reviewer has been incorporated in the revised version of the manuscript.
Comment: Results:   Explanatory Notes: It would be helpful to include some explanatory notes or footnotes for the abbreviations and terms used in the table, such as "HPLP II" (Health-Promoting Lifestyle Profile II), to ensure that readers understand the meaning of these terms. Interpretation of Results: While the table presents a lot of numerical data, it's essential to provide some understanding of the key findings. For example, what do the correlations in Table 2 imply? How do they relate to the study's objectives or hypotheses? Providing some context and interpretation can help readers understand the significance of the results. Discussion of Disease Categories: In the section discussing disease categories, you may want to elaborate on why certain diseases are more prevalent among the student population. Are there any potential factors or trends that could explain these findings? Discussion of Subscale Scores: In the section where you present the subscale scores of the Health-Promoting Lifestyle Profile (HPLP II), you could provide some context or discussion about why specific subscales have higher or lower scores. Are there any implications for the student’s overall well-being or health behaviours? Multiple Linear Regression Interpretation: When presenting the results of multiple linear regression in Tables 4 and 5, provide interpretations of the coefficients. Explain what a positive or negative coefficient means in the context of the study and how it relates to the predictor variables. Statistical Significance: Indicate which results are statistically significant (e.g., using asterisks or symbols) to highlight the most important findings. Conclusion of the Results Section: Summarize the main findings of the results section before moving on to the discussion. This will help readers grasp the key takeaways from this section.
Reply: The suggestions and recommendations of the reviewer has been incorporated in the revised version of the manuscript.
Comment: Discussion Introduction and Context Setting: Start the discussion section with a brief recap of the study's main objectives and key findings to provide context for readers who may have not read the entire article. Mention the significance of the study and its potential implications for public health or university policies. Structure and Flow: Organize the discussion logically by discussing each factor or finding one at a time and then drawing connections between them. Use subheadings to delineate different aspects of the discussion, making it easier for readers to follow your arguments. Interpretation of Findings: Provide more in-depth interpretation and analysis of the study's findings. For example, discuss why certain factors, such as mental health or disease presence, are associated with lower health promoting behaviours. What underlying mechanisms might explain these associations? Consider discussing the potential reasons behind the variation in lifestyle behaviours among different subgroups of students (e.g., cultural factors, societal norms, or campus environments). Comparison with Prior Research: Discuss how your study's findings compare and contrast with previous research in this area, both within and outside Saudi Arabia. Mention any trends or changes observed compared to studies conducted before and after the COVID-19 pandemic, as it may have influenced lifestyle behaviours.
Reply: The suggestions and recommendations of the reviewer has been incorporated in the revised version of the manuscript.
Comment: Limitations and Future Directions: Acknowledge the limitations of your study, such as potential biases or limitations in data collection methods. This demonstrates transparency. Suggest directions for future research. What questions remain unanswered, and how could future studies address them?
Reply: The suggestions and recommendations of the reviewer has been incorporated in the revised version of the manuscript.
Comment: Practical Implications: Discuss the practical implications of your findings. What can universities or policymakers do to promote healthier lifestyles among students, especially regarding observed sedentary behaviour and sleep disorders? Consider addressing potential interventions or strategies that could improve the health-promoting activities of university students. Conclusion: Summarize the key takeaways from your discussion, emphasizing the most important findings and their implications. Avoid introducing new information or data in the conclusion; it should primarily be a summary.
Reply: The suggestions and recommendations of the reviewer has been incorporated in the revised version of the manuscript.
Comment: Citations: Ensure you appropriately cite previous research when comparing or drawing on prior findings. Recommendations: Specificity and Clarity: While you provide some valuable recommendations, consider making them more specific and actionable. For example, instead of suggesting "promoting physical activity," specify the types of activities or programs that could be introduced, such as yoga classes, fitness challenges, or intramural sports.
Reply: The suggestions and recommendations of the reviewer has been incorporated in the revised version of the manuscript.
Comment: Prioritization: Please prioritize your recommendations based on the findings. For instance, if sedentary behaviour is a more significant concern than sleep patterns, emphasize interventions to reduce sedentary behaviour first.
Reply: The suggestions and recommendations of the reviewer has been incorporated in the revised version of the manuscript.
Comment: Evidence-Based Interventions: Support your recommendations with evidence from existing research or best practices in promoting health and well-being among university students. This adds credibility to your suggestions.
Reply: The suggestions and recommendations of the reviewer has been incorporated in the revised version of the manuscript.
Comment: Inclusivity: Consider addressing the diverse needs of university students. Not all students have the same lifestyle patterns or require the same interventions. Mention strategies for engaging various subgroups, such as international students, students with disabilities, or those from different cultural backgrounds.
Reply: The suggestions and recommendations of the reviewer has been incorporated in the revised version of the manuscript.
Comment: Feasibility: Discuss the feasibility and potential challenges of implementing these recommendations in a university setting. This can help institutions assess the practicality of adopting such interventions.
Reply: The suggestions and recommendations of the reviewer has been incorporated in the revised version of the manuscript.
Comment: Limitations: Causal Inference: In the "Limitations" section, you correctly point out the limitations of the cross-sectional design. However, you can also suggest how future research, such as longitudinal studies, could address these limitations and provide insights into causal relationships.
Reply: The suggestions and recommendations of the reviewer has been incorporated in the revised version of the manuscript.
Comment: Socially Acceptable Responses: Acknowledge that self-reported data can be subject to biases, but also mention how the study attempted to mitigate this bias (e.g., by including large sample size) and suggest potential ways to improve data collection in future studies, such as using objective measurements or combining self-reporting with other data sources.
Reply: The suggestions and recommendations of the reviewer has been incorporated in the revised version of the manuscript.
Comment: Generalizability: While you mention that the study was conducted in a single university in Riyadh, Saudi Arabia, you can also discuss the potential cultural or regional factors that might influence the findings. This can help readers understand the context better and assess whether the results could apply to other settings.
Reply: The suggestions and recommendations of the reviewer has been incorporated in the revised version of the manuscript.
Comment: Ethical Considerations: Briefly mention any ethical considerations or limitations related to the study, such as informed consent, privacy, or potential participant biases.
Reply: The suggestions and recommendations of the reviewer has been incorporated in the revised version of the manuscript.
Comment: Conclusion Summarize Key Findings: In this section, it's crucial to provide a concise summary of the main findings of your study. Mention the critical lifestyle behaviours identified as areas of concern among university students. This will help readers quickly grasp the most important takeaways from your research.
Reply: The suggestions and recommendations of the reviewer has been incorporated in the revised version of the manuscript.
Comment: Link to Research Questions/Hypotheses: If your study had specific research questions or hypotheses, briefly reiterate them and then state whether they were supported by your findings. This provides a clear link between your research objectives and the results.
Reply: The suggestions and recommendations of the reviewer has been incorporated in the revised version of the manuscript.
Comment: Practical Implications: Discuss the practical implications of your findings. How can the insights gained from this study be applied in real-world scenarios? How can policymakers and university administrators use this information to develop effective interventions?
Reply: The suggestions and recommendations of the reviewer has been incorporated in the revised version of the manuscript.
Comment: Recommendations for Future Research: Expand on the need for future research mentioned in the conclusion. Provide more specific suggestions for future studies, particularly those involving longitudinal analysis, to assess the effectiveness of interventions.
Reply: The suggestions and recommendations of the reviewer has been incorporated in the revised version of the manuscript.
Comment: Concluding Statement: End the conclusion with a strong concluding statement that highlights the broader significance of your study. Explain why addressing lifestyle behaviours among university students is essential, both from a public health perspective and for the student’s well-being.
Reply: The suggestions and recommendations of the reviewer has been incorporated in the revised version of the manuscript.
Comment: Avoid Repetition: The phrase "insinuate that there is a need for health promotion interventions" could be replaced with a more direct statement. Additionally, phrases such as "Further, the results of this study will undoubtedly assist policymakers and university administrators" could be made more concise and to the point.
Reply: The suggestions and recommendations of the reviewer has been incorporated in the revised version of the manuscript.

Reviewer 3 Report
Comments and Suggestions for Authors
The manuscript by Alothman and colleagues conducted a cross-sectional study to evaluate the factors associated with lifestyle behaviors among the students at the Princess Nourah Bint Abdulrahman University in Riyadh Saudi Arabia. Overall, this is an interesting study, and the manuscript is well-written. However, the statistical methods and the interpretations of their study results must be improved.
- Since this is a cross-sectional study, the temporal order of some independent and dependent variables is unclear. Thus, a significant association only shows correlation instead of causation, and it's misleading to use the word "prediction" when interpreting the results. For example, the word “predictors” was used in the titles of Tables 4 and 5. In discussion, the authors concluded that “mental health conditions exerted a notable adverse influence on various subclasses of the HPLP II (P<0.05 in all cases)”. It’s also possible that a healthier lifestyle leads to better mental health. Since it’s hard to tell which happens first, “association” or “correlation” are more appropriate to interpret the study results.
- You mentioned all the students who attended PNU during 2021 and 2022 academic years were included in the study. It’s unclear if all of them respond to survey questions. Are there any missing data in this study? How many students were included in the final multiple linear regression?
- It’s unclear what the variable selection process is for the multiple linear regression models. Some independent variables that were not correlated with the dependent variables in the ANOVA tests were selected into the linear regression models. For example, college was not associated with HPLP or sleep in Table 3, but was included in the linear regression in Table 4. It has five categories, including college in your model might reduce your statistical power. The R squares are pretty small in Tables 4 and 5. Conducting a variable selection could help improve your model fit. In addition, have you done model diagnoses of the final regression models? Any outliers?
Author Response
Reply to the comments of Reviewer 3
Comment: The manuscript by Alothman and colleagues conducted a cross-sectional study to evaluate the factors associated with lifestyle behaviors among the students at the Princess Nourah Bint Abdulrahman University in Riyadh Saudi Arabia. Overall, this is an interesting study, and the manuscript is well-written. However, the statistical methods and the interpretations of their study results must be improved.
Reply: The authors are grateful for the appreciation and encouragement received from Reviewer 3.
- Comment: Since this is a cross-sectional study, the temporal order of some independent and dependent variables is unclear. Thus, a significant association only shows correlation instead of causation, and it's misleading to use the word "prediction" when interpreting the results. For example, the word “predictors” was used in the titles of Tables 4 and 5.
Reply: The authors have made changes as per their understanding to tables 4 and 5.
- Comment: In discussion, the authors concluded that “mental health conditions exerted a notable adverse influence on various subclasses of the HPLP II (P<0.05 in all cases)”. It’s also possible that a healthier lifestyle leads to better mental health. Since it’s hard to tell which happens first, “association” or “correlation” are more appropriate to interpret the study results.
Reply: The authors appreciate the genuine and considerable feedback by the expert reviewer and accept the recommendations. The changes were made accordingly.
- Comment: You mentioned all the students who attended PNU during 2021 and 2022 academic years were included in the study. It’s unclear if all of them respond to survey questions. Are there any missing data in this study? How many students were included in the final multiple linear regression?
Reply: The students who attended the university in the session 2021-2022 were invited to respond to this cross-sectional study, but not all of them participated. This has been added to the participants sections of the manuscript. There was no missing data, all the students (1112) responded the complete questionnaire. All the students were included in the final multiple linear regression.
- Comment: It’s unclear what the variable selection process is for the multiple linear regression models. Some independent variables that were not correlated with the dependent variables in the ANOVA tests were selected into the linear regression models. For example, college was not associated with HPLP or sleep in Table 3, but was included in the linear regression in Table 4. It has five categories, including college in your model might reduce your statistical power. The R squares are pretty small in Tables 4 and 5. Conducting a variable selection could help improve your model fit. In addition, have you done model diagnoses of the final regression models? Any outliers?
Reply: The suggestions and recommendations of the reviewer has been incorporated in the revised version of the manuscript.

Reviewer 4 Report
Comments and Suggestions for Authors
Please immediately revise the manuscript according to the suggestions for improvement that I wrote in your manuscript!

Please immediately revise the manuscript according to the suggestions for improvement that I wrote in your manuscript!
Author Response
Reply to the comments of Reviewer 4
Comment: Please immediately revise the manuscript according to the suggestions for improvement that I wrote in your manuscript!
Comment: 1. If seen from the title raised, this research has not really contributed to science, because it only explores factors without any in-depth meaning.
Reply: We appreciate the reviewer’s feedback; however, we are surprised to see the comment, the study has not been conducted in Saudi population with the variables included in our study.
Comment: 2. The method description is not strong enough! The reasons for choosing this research have not been explained, there has been no explanation regarding this type of research, there has been no explanation regarding sampling techniques and sampling characteristics, there has been no explanation regarding the characteristics of instruments and data collection procedures, and there has been no explanation regarding the data analysis techniques used.
Reply: The reasons for conducting this study have been inserted from line 61-74. The sampling technique has also been included at line number 114-115. Characteristics of the instrument has been included from line 116-135. The explanation of data analysis has been described from line 136-148.
Comment: 3. Why does the abstract description of findings only duplicate the findings in the results section? This just replicates without any meaning. The findings obtained are only descriptive and inferential in nature, there is no further meaning to this research!
Reply: The suggested changes have been made in the abstract.
Comment 4. The meaning of these inferential quantitative findings has not been interpreted logically and there has been no explanation of the conclusions and implications!
Reply: The suggested changes have been made in the abstract and the interpretations have been included.
Comment: 6. Keywords are sorted in alphabetical order and a maximum of only 6 keywords are written!
Reply: The authors have made the suggested changes.
Comment: 7. Please provide an explanation of the scientific reasons underlying this research! This is because the description of the gap between facts and theory that should occur is still shallow. The contents of the introduction section that you describe are only previous research claims without a strong argument.
Reply: The changes have been made and inserted in the manuscript from line number 93.
Comment: 8. Please add a research question at the end of this introduction section!
Reply: The authors have made the suggested changes.
Comment: 9. Whose reference did you use for the cross-sectional study?
Reply: The reference has been added.
Comment: 10. Why only this academic year? Give a scientific reason!
Reply: The students who attended the university in the session 2021-2022 were invited to respond to this cross-sectional study, but not all of them participated. This year was selected because the project was meant to be completed in a span of 12 months and since it was approved in the year 2021, we had only one option left, i.e., to select the participants from the current session. This has been added to the participants sections of the manuscript.
Comment: 11. Please add a review of the sampling technique used in this research! Also add reasons why the sample can be involved in this research! Also describe the characteristics of the sample in this study!
Reply: The authors have made the suggested changes.
Comment: 12. Please arrange the results section into several sub-chapters whose contents and sequence are adjusted to the research questions! Please, for each table included in the manuscript, you can sort the important content only, and the unimportant content can be removed from the table so that it doesn't take up too many pages!
Reply: The authors believe they have provided optimal information that is needed.
Comment: 13. table 1. The table is too long, please summarize it again or just move it to the appendix.
Reply: The authors believe they have provided optimal information that is needed.
Comment: 14. table 3. The table is too long, please summarize it again or just move it to the appendix.
Reply: The authors believe they have provided optimal information that is needed.
Comment: 15. table 4. The table is too long, please summarize it again or just move it to the appendix.
Reply: The authors believe they have provided optimal information that is needed.
Comment: 16. table 5. The table is too long, please summarize it again or just move it to the appendix.
Reply: The authors believe they have provided optimal information that is needed.
Comment: 17. (conclusion) Add an explanation of the meaning of these findings and their implications!
Reply: The authors have made the suggested changes.

Round 2
Reviewer 2 Report
Comments and Suggestions for Authors
As the authors have not implemented the previously recommended changes, I hope they will review and address the previous comments accordingly.
Comments on the Quality of English LanguageModerate editing of English language required
Author Response
As the authors have not implemented the previously recommended changes, I hope they will review and address the previous comments accordingly.
We are sorry to hear you are not satisfied with our 1st round of responses. However, we hope we had clarified and enhanced our response inn this round of revision.
Abstract
- Clarity of Purpose and Scope: The abstract should clearly state the study’s primary objective or research question. What specifically were the researchers trying to investigate? In this case, it's about lifestyle behaviors among university students, but it would be helpful to be more explicit about the main aim.
Reply: Purpose was clarified as following “This study examined lifestyle behaviors and their associations with socio-demographics, and health determinants (comorbidities and pain) in a Saudi university students’ during the academic year 2021 and 2022”
- Methodology: The abstract briefly mentions using Pearson's correlation and multiple linear regression, but it should provide more detail about the methods used in the study. Saying the sample size, data collection methods, and any significant statistical tests or measures would be beneficial for readers to understand how the study was conducted.
Reply: Methos were revised as follow “All the university students received the study invitation via university emails to complete an online questionnaire. The questionnaire included four sections (socio-demographics, health-related information, desired health promotion activities, and lifestyle behavior assessment via the Health-Promoting Lifestyle Profile II (HPLP-II). Associations between study variables were assessed using Pearson's correlation and multiple linear regression.”
- Key Findings: While the abstract mentions some correlations and results, it could be more explicit about the key findings. For example, instead of just saying positive or negative correlations, briefly summarize the main findings related to lifestyle behaviours and their associations with socio-demographics, comorbidities, and pain. Use specific values or percentages to provide a clearer picture of the results.
Reply: Key findings were revised to match study objective mentioned in the abstract.
- Significance and Implications: The abstract should include a sentence or two discussing the significance of the study and its potential implications. Why are these findings significant, and how might they be used to benefit students or the university community? What are the broader implications for health promotion interventions?
Reply: Significance and Implications were revised and broader implications were added (lines 43-44)
- Limitations: It's essential to acknowledge any limitations of the study in the abstract. Were there any limitations in the data, methodology, or sample that could affect the validity of the findings? Being transparent about limitations adds credibility to the research.
Reply: Limitations were added (lines 45-46)
- Conclusion: Conclude the abstract by summarizing the main takeaway or message of the study. What should readers remember after reading the abstract? In this case, it could be a clear statement about the need for health promotion interventions among Saudi university students based on the study's findings.
Reply: Conclusion sentence was added (line 44-47)
- Formatting: Ensure that the abstract is well-structured and follows a logical sequence. Use clear headings or subheadings if necessary to organize the content.
Reply: We paid special attention to match the journal formatting as recommended.
Introduction
- Clearly Define the Research Problem: The introduction should start by clearly defining the research problem or gap in the existing literature. What specific aspect of lifestyle behaviours among university students is the study addressing? Is there a particular research question or hypothesis the study aims to answer? Providing a clear research focus will help readers understand the significance of the study.
Reply: 1. In the 1st paragraph we have included Different populations based on geographical location, age, disease status, or any common grouping factor exhibit different trends and norms of each behavior, thus, it is imperative to assess above mentioned lifestyle behaviors and theirs related associates within each specific population to generate relevant health data.” Sentence to allude readers for the study objectives early on.
- Provide a Rationale for the Study: Explain why investigating lifestyle behaviours among university students is essential. What are the potential health implications, especially for young adults? Is there any particular reason why Princess Nourah Bint Abdulrahman University (PNU) students were chosen for this study?
Reply: 2. Revision presented in Lines 93-111 address this point
- Review of Relevant Literature: While the introduction mentions some studies from Turkey, Germany, and Saudi Arabia, providing a more comprehensive review of relevant literature on lifestyle behaviours among university students would be helpful. This could include statistics, trends, and key findings from previous research to establish a stronger foundation for the study.
Reply: 3. This point was addressed as “However, there is no comprehensive reviews predicting prevalence and direct effect of such behaviors among university students. The best evidence available comes from individual country specific studies.” And including Saudi specific numbers “These studies indicated that at least 30% to 85% of the students were physically inactive or sedentary, up to 95% did not get enough sleep, up to 70% were stressed, and from 50% to 90% did not engage in healthy eating habits.”
- Justify the Need for a Holistic Approach: The introduction mentions that lifestyle behaviours have been investigated as separate domains. Still, it must provide a clear rationale for why a holistic approach, considering multiple lifestyle factors together, is essential. Explain the potential benefits of understanding the clustering effect of these behaviours and how it can contribute to a more comprehensive assessment of students' health.
Reply: 4. New reference was added to support. “This clustering pattern of different lifestyle behaviors could potentially affect the efficacy and effectiveness of lifestyle intervention especially in university students.”
- Highlight the Study's Objectives: Clearly state the specific objectives of the study in the introduction. What are the researchers trying to achieve by examining lifestyle behaviours, comorbidities, and pain among university students at PNU? This will give readers a sense of what to expect from the study.
Reply: 5. This point was addressed in lines 110-117
- Transition to the Methodology: The introduction could end with a smooth transition to the methodology section, briefly mentioning how the study was conducted, the data collection methods used, and the population under investigation.
Reply: 6. In attempt to transition smoothly to methods we adjust the last sentence in the introduction to the following “Therefore, this study aimed to evaluate self-reported lifestyle behaviors and their associations with socio-demographics, comorbidities, and pain among university students at PNU utilizing a cross-sectional study design.”
Methodology
- Study Design:While the section mentions that a cross-sectional study was conducted at Princess Nourah Bint Abdulrahman University (PNU), it would be helpful to briefly explain why a cross-sectional design was chosen for this particular study. Mention the advantages of this design for studying lifestyle behaviours among university students.
Reply: 1. Study design section was revised accordingly
- Participants and Recruitment: Provide information about the total number of students at PNU during the 2021 and 2022 academic years. This will help readers understand the population size from which the sample was drawn.
Reply: 2. Students total was approximately twenty-seven thousand. This information was added.
- Ethical Considerations and Consent: Mention the specific ethical considerations that were taken into account during the study. For example, did the researchers consider informed consent, privacy, and confidentiality issues? Explain how these ethical considerations were addressed in the study design.
Reply: 3. Ethical Considerations and Consent section was revised to clarify the consent process.
- Sample Size Calculation: Explain the rationale behind choosing a 98% confidence level and a 5% margin of error for sample size calculation. Why were these specific values determined, and how do they relate to the research objectives? Also, please explain the clustered design effect and why it was considered in the sample size calculation.
Reply: 4. We believe wider CI of 98% will help us get better estimate to multiple variables assessed. Because we wanted to compare our data at the collages level we wanted to account for the clustering affect too. All in all, were used sample size calculation to calculate the minimal number of students needed to assess the study objective.
- Study Questionnaire: Provide more details about the content of the study questionnaire. What questions were included in each part of the questionnaire related to age, comorbidities, pain, health promotion activities, and lifestyle behaviours? This will give readers a better understanding of the data collected.
Reply: 5. Study questionnaire section was revised to include more details about the questionnaire and why we chose it.
- Health-Promoting Lifestyle Profile II (HPLP-II):Explain why the HPLP-II instrument was chosen for assessing lifestyle behaviours. Briefly describe the six sub-scales and how they contribute to determining health-promoting behaviour. Additionally, clarify why the mean of the sub-scales was used rather than the total scores.
Reply: 6. Study questionnaire section was revised to include more details about the questionnaire and why we chose it. Further, both the total and mean of scale were used as shown in the results section. We corrected the information
- Data and Statistical Analyses: While the section mentions the statistical software used (JMP Pro, version 14.2), providing a brief rationale for why this software was chosen for data analysis would be beneficial. Additionally, consider providing some information on the specific statistical tests used, especially for the independent samples, analysis of variance (ANOVA), Pearson's correlation coefficient analysis, and multiple linear regression. This will help readers understand how the data were analyzed.
Reply: 7. JMP was chosen because it is the available software for the researchers, we are not aware of any significant difference between JMP or SPSS or GraphPad software.
Results
- Explanatory Notes: It would be helpful to include some explanatory notes or footnotes for the abbreviations and terms used in the table, such as "HPLP II" (Health-Promoting Lifestyle Profile II), to ensure that readers understand the meaning of these terms.
Reply: 1. Explanatory Notes were added for all tables.
- Interpretation of Results: While the table presents a lot of numerical data, it's essential to provide some understanding of the key findings. For example, what do the correlations in Table 2 imply? How do they relate to the study's objectives or hypotheses? Providing some context and interpretation can help readers understand the significance of the results.
Reply: 2. Table 2 was deleted as it does not add relevant information for the current study objectives
- Discussion of Disease Categories: In the section discussing disease categories, you may want to elaborate on why certain diseases are more prevalent among the student population. Are there any potential factors or trends that could explain these findings?
Reply: 3. No diseases were out of expected, secondary analysis revealed no associations
- Discussion of Subscale Scores: In the section where you present the subscale scores of the Health-Promoting Lifestyle Profile (HPLP II), you could provide some context or discussion about why specific subscales have higher or lower scores. Are there any implications for the student’s overall well-being or health behaviours?
Reply: 4. Discussion of subscales were included in the discussion section
- Multiple Linear Regression Interpretation: When presenting the results of multiple linear regression in Tables 4 and 5, provide interpretations of the coefficients. Explain what a positive or negative coefficient means in the context of the study and how it relates to the predictor variables.
Reply: 5. Revision was made to illustrate these relations clearly
- Statistical Significance: Indicate which results are statistically significant (e.g., using asterisks or symbols) to highlight the most important findings.
Reply: 6. statistically significant results were highlighted in the tables
- Conclusion of the Results Section: Summarize the main findings of the results section before moving on to the discussion. This will help readers grasp the key takeaways from this section.
Reply: 7. main findings were summarized in the 1st section in the discussion section as recommended.
Discussion
- Introduction and Context Setting:
- Start the discussion section with a brief recap of the study's main objectives and key findings to provide context for readers who may have not read the entire article.
Reply: 1. a. we have started the discussion with “This cross-sectional study looked at lifestyle choices made by PNU students in various colleges throughout the academic year 2021–2022, and their relationships with sociodemographic characteristics, comorbid conditions, and pain status” to help readers to recap.
- Mention the significance of the study and its potential implications for public health or university policies.
Reply: 1.b. statement related to significance of the study and its potential implications for public health or university policies were added
- Structure and Flow:
- Organize the discussion logically by discussing each factor or finding one at a time and then drawing connections between them.
- Use subheadings to delineate different aspects of the discussion, making it easier for readers to follow your arguments.
Reply: 2. Organization were revised. We thank the reviewer for their suggestion of utilizing subheading, however, we think traditional discussion section without any subheading is the best fit to give the readers the best experience
- Interpretation of Findings:
- Provide more in-depth interpretation and analysis of the study's findings. For example, discuss why certain factors, such as mental health or disease presence, are associated with lower health-promoting behaviours. What underlying mechanisms might explain these associations?
- Consider discussing the potential reasons behind the variation in lifestyle behaviours among different subgroups of students (e.g., cultural factors, societal norms, or campus environments).
Reply: Discussion the relationships in-depth as suggested could mislead the readers and infer cause and effect of these related factors. However, more discussion points were added.
- Comparison with Prior Research:
- Discuss how your study's findings compare and contrast with previous research in this area, both within and outside Saudi Arabia.
Reply: This point was covered throughout the discussion section
- Mention any trends or changes observed compared to studies conducted before and after the COVID-19 pandemic, as it may have influenced lifestyle behaviours.
Reply: This point was covered throughout the discussion section
- Limitations and Future Directions:
- Acknowledge the limitations of your study, such as potential biases or limitations in data collection methods. This demonstrates transparency.
Reply: Limitations were updated accordingly
- Suggest directions for future research. What questions remain unanswered, and how could future studies address them?
Reply: Limitations were updated accordingly
- Practical Implications:
- Discuss the practical implications of your findings. What can universities or policymakers do to promote healthier lifestyles among students, especially regarding observed sedentary behaviour and sleep disorders?
Reply: The suggestion was considered, and changes were made accordingly in the manuscript’s discussion section.
- Consider addressing potential interventions or strategies that could improve the health-promoting activities of university students.
Reply: The recommended action was considered and changes were made accordingly in the discussion section of the manuscript.
- Conclusion:
- Summarize the key takeaways from your discussion, emphasizing the most important findings and their implications.
Reply: All the points were summarized in the conclusion section with the suggested emphasis.
- Avoid introducing new information or data in the conclusion; it should primarily be a summary.
Reply: No new information or data was included in the conclusion section as suggested by the erudite reviewer.
- Citations:
- Ensure you appropriately cite previous research when comparing or drawing on prior findings.
Reply: All the citations were double checked and proper use of citation was take care of.

Reviewer 3 Report
Comments and Suggestions for Authors
The authors have adequately addressed my comments.
Author Response
|
The authors have adequately addressed my comments. |
Thank you |

Reviewer 4 Report
Comments and Suggestions for Authors
Please immediately revise your manuscript according to the suggestions for improvement that I provided in your manuscript!

Please immediately revise your manuscript according to the suggestions for improvement that I provided in your manuscript!
Author Response
Response to reviewers – round two. We extremely appreciate the editor and reviewers for reviewing our manuscript. We have addressed the reviewers’ comments as described below.
|
Reviewer 4 |
|
|
Please immediately revise your manuscript according to the suggestions for improvement that I provided in your manuscript! |
Revision was made as requested |
|
Please separate the discussion section from the conclusion section. Clarify the description of the conclusion section! |
In the revised manuscript you can find the conclusion section right below the dissection section as recommended, and it was clarified as follow:
|
|
Write the conclusion section right below the discussion section! |
In the revised manuscript you can find the conclusion section right below the dissection section as recommended |
|
Please add some references below ! Kistoro, H. C. A., Kartow agiran, B., Naim, N., .. (2020). Islamophobia in education: perceptions on the wear of veil/ nigab in higher education. Indonesian Journal of Islam and Muslim Societies, 10 (2), 227-246. 3. |
We find the suggest reference intriguing. However, we fail to see the connection between it and the current study. |

Round 3
Reviewer 2 Report
Comments and Suggestions for Authors
Dear Authors,
We want to acknowledge that there have been some improvements based on the recommendations provided.
Best Regards
Comments on the Quality of English Languagea thorough reading to check for minor spelling errors is recommended